# Semi-Structured Object Sequence Encoders

**Rudra Murthy V[1], Riyaz Bhat[1], Chulaka Gunasekara[1], Siva Sankalp Patel[1]**
**Hui Wan[1], Tejas Indulal Dhamecha[2]\*, Danish Contractor[1], Marina Danilevsky[1]**

[1]IBM Research AI
[2]Microsoft India Development Center
rmurthyv@in.ibm.com,{riyaz.bhat,chulaka.gunasekara,siva.sankalp.patel}@ibm.com,
hwan@us.ibm.com, tdhamecha@microsoft.com,
danish.contractor@ibm.com, mdanile@us.ibm.com

## Abstract

In this paper we explore the task of modeling semi-structured object sequences; in particular, we focus our attention on the problem of developing a structure-aware input representation for such sequences. Examples of such data include user activity on websites, machine logs, and many others. This type of data is often represented as a sequence of sets of key-value pairs over time and can present modeling challenges due to an ever-increasing sequence length thereby affecting the quality of the representation. We propose a two-part approach, which first considers each key independently and encodes a representation of its values over time; we then self-attend over these value-aware key representations to accomplish a downstream task. This allows us to learn better representation while being able to operate on longer object sequences than existing methods. We introduce a novel shared-attention-head architecture between the two modules and present an innovative training schedule that interleaves the training of both modules with shared weights for some attention heads.[1] Our experiments on multiple prediction tasks using real-world data demonstrate that our approach outperforms a unified network with hierarchical encoding, as well as other methods including a *record-centric* representation and a *flattened* representation of the sequence.

## 1 Introduction

Semi-structured object sequences comprise a significant portion of the myriad of data created daily. This data usually has a temporal aspect, with the data created sequentially and representing events happening in some order. More generally, the data is a sequence of structured objects, each represented by a set of key-value pairs that encode the attributes of the object (Figure 1(a)). Examples include recordings of user interactions with websites, logs of machine activity, shopping decisions made by consumers, and many more (Figure 1(b)). The data is usually stored in semi-structured formats such as JSONs, or tabular forms.

In this paper, we explore the task of modeling semi-structured object sequences; in particular, we focus our attention on the problem of developing a structure-aware input representation for such sequences. If we think of the parallel to natural language data, we could treat each sentence of a text (Figure 1(c)) akin to the set of key-value pairs at a particular time step.

**The challenge of sequence length:** A trivial method of representing such sequences would be to *flatten* each structured object and view its constituents as individual *words* for tokenization in natural language (Figure 1 (d)).[2] However, this causes the sequence length to become extremely large (thousands of tokens) when operating on real-word semi-structured sequences. For instance, in our study of semi-structured objects from user-interaction sessions on software from a large cloud-based service provider, we found these objects could contain 60 fields on average. The values of these fields include timestamps, identifiers, log messages, etc., with an average of 5 words each. A session length of 15 minutes results in 105 such session objects, amounting to nearly $31,500$ words which would further increase the sequence length after sub-words are created.

**Record-centric representation:** In contrast, to flattening structured objects, one could also create *record-centric* representations (Figure 1(f)) where the set of key-value representations at each time step are considered in sequence. As our experiments in Section 3.3 demonstrate, such represen-

---

\*Work carried out when the author was an IBM employee
[1]Code at https://github.com/murthyrudra/SemiStructuredEncoders

[2]with markers to indicate boundaries for each structured object

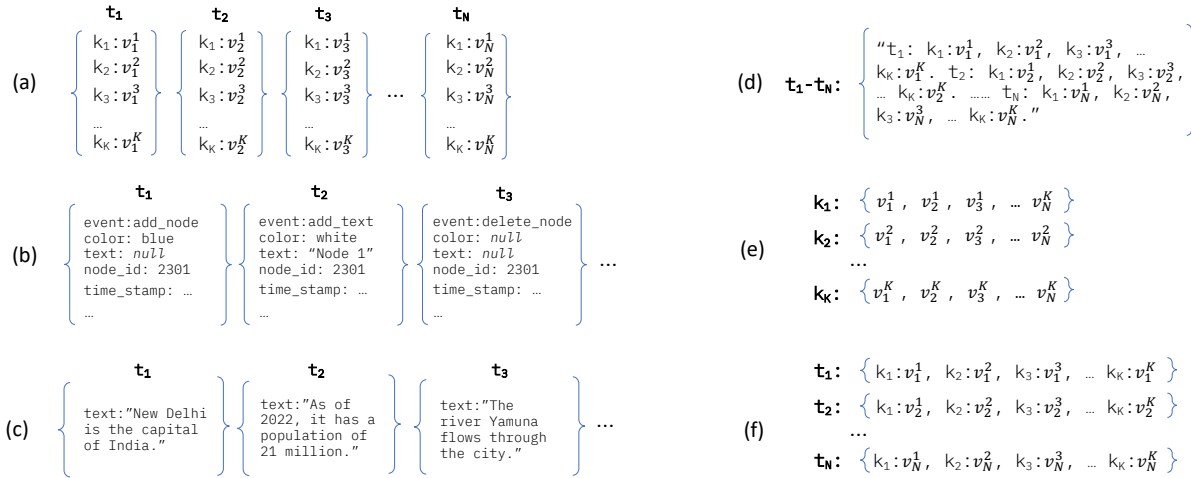

Figure 1: Semi-Structured Object Sequences: (a) Generic representation of a sequence of semi-structured objects, consisting of multiple key-value pairs at time steps $t_1 \dots t_N$. (b) Example object: a sequence of events triggered by the use of a graphical user interface. (c) Viewing a text paragraph as a sequence of sentences. (d) Encoding the example object in (a) by *flattening*. (e) Encoding (a) by encoding a representation of the values for each key (to be followed by a key aggregation step, not shown). (f) Encoding (a) using a *record-centric* representation for each time step.

tations do not work well when operating on long sequences. We hypothesize that this could be because the record-representation does not adequately model the dependencies between the constituent values of the same key across different time steps (records).

**Key-centric representation:** To address these challenges we use a modular, two-part hierarchical encoding strategy. First, we decompose the sequence of semi-structured objects into independent sets of sequences based on keys. This allows us to consider each key separately (Figure 1(e)), and encode a representation of how the values of that key evolve over time (we refer to this as Temporal Value Modeling – TVM). This may be achieved using any encoder.[3] We then self-attend over the set of the key encodings to create a representation of the entire structured object sequence (referred to as Key Aggregation – KA).

**Advantages:** This key-centric perspective of encoding semi-structured sequences has many advantages as compared to *flattening* and *record-centric representations* (Figure 1(f)).

Decoupling the keys allows us to support an arbitrary number of keys[4] since each key-sequence is encoded independently. So, the key-representations created during Temporal Value Modeling can sup-

port longer sequences than what would have been impossible with flattening (due to memory constraints). Moreover, this encoding strategy also accommodates input sequences that may be considered non-structured – e.g, natural language text as sequences of words in sentences (Figure 1(c)). Specifically, if we consider a sequence of structured objects where each structured object consists of only one key, whose values contain a *sentence*, then our TVM effectively encodes the text sequence using whichever encoder has been employed.[5]

The decoupling of keys and the use of two independent encoders - TVM for value-aware key representations, and KA for aggregating key-representations for a downstream task - requires that information be shared between the two networks so that the key-representations generated by TVM can be optimized for downstream tasks via the KA. To facilitate this, we share a few sets of attention heads between the two networks. First, we pre-train the TVM network with shared attention heads in place. We then use the frozen representations from this network to initialize the KA network, which has its own untrained attention heads, and the shared attention heads from the TVM network as part of its trainable parameters. We utilize a training schedule that interleaves the training of both modules to iteratively train them. Doing so allows the TVM and KA modules to create richer

---

[3]We use BERT and Longformer in our experiments

[4]Real-world data can have hundreds of keys in each object.

[5]The key aggregation step, in this case, is redundant.

representations of keys, informed by their importance for the downstream task. We find that this novel iterative two part-training results in better performance compared to a unified network with hierarchical encoding (with no attention-head sharing) as well as other methods, that either use a *flattened* representation or a *record-centric* representation of the sequence (Veličković et al., 2018; Mizrachi and Levin, 2019; de Souza Pereira Moreira et al., 2021; Padhi et al., 2021).

**Contributions:** Our work addresses the challenges of encoding semi-structured object sequences: (i) we propose a two-part approach that separately encodes the evolution of the values for each key, followed by aggregation over key-representations to accomplish downstream tasks; (ii) we present a novel approach for sharing attention heads between the components; (iii) we compare our approach against baselines such as sequence flattening, joint encoding, and record-centric sequence representations; and (iv) we present detailed experiments on several datasets and tasks to demonstrate the advantages of our approach.

## 2 Modeling

We now describe our approach for modeling key-value semi-structured object sequences.

Let a sequence of semi-structured objects be denoted as $\mathcal{J} = [J_1, J_2, J_3, \ldots, J_N]$ corresponding to $N$ time steps. Further, let $J_i = \{k_1 \colon v_i^1, k_2 \colon v_i^2, \ldots, k_j \colon v_i^j, \ldots, k_K \colon v_i^K\}$ denote a structured object $J_i$, containing $K$ key-value pairs $< k_j \colon v_i^j >, j = 1 \ldots K$. The goal of our modeling is to learn a representation of a sequence of structured objects $\mathcal{J}$; and subsequently, learn $f \colon \mathrm{Embd}(\mathcal{J}) \to \{1, 2, \ldots, C\}$ for an end task, such as a $C$-way classification task.

We develop a modular two-part modeling strategy to represent a sequence of structured objects.

1. The first module, called the Temporal Value Modeler (TVM), is used to learn a combined representation (referred to as the *key-representations*) for the different values that each key takes in the sequence.

2. The second module, called the Key-Aggregator (KA), uses the *key-representations* corresponding to each key, to create an overall representation for $\mathcal{J}$.

**Temporal Value Modeling**: Let $k$ be a key from the universe of all the keys $\mathcal{K}$ in the sequence. Then,

for each key $k$ we encode the value-aware key-representation $V_k$, by considering the value of the key $k$ at each timestamp, as a sequence. Formally, $V_k$ is given by:

$$V_k = [\texttt{CLS}] \, k [\texttt{VAL\_SEP}] v_1^k \, [\texttt{VAL\_SEP}] \, v_2^k \, [\texttt{VAL\_SEP}] \ldots v_N^k \tag{1}$$

where [VAL_SEP] and [CLS] are special tokens. Note that each value $v_j^k$ for a key $k$ at time step $j$ can itself consist of many tokens and those have not been shown for ease of presentation. With any choice of a transformer-based (Vaswani et al., 2017) language encoder (*TextEncoder – TE*), an embedding for $V_k$, termed the *key-representation* (KR), can be obtained as:

$$\mathrm{KR}_k = TextEncoder(V_k) \tag{2}$$

We use the output embedding representation at the first position (corresponding to [CLS] ) as the *key-representation*. It is easy to see that this formulation allows us to accommodate the modeling of natural language text as in Figure 1 (c). For illustration, if the $TextEncoder$ is based on BERT (Devlin et al., 2019), Eq. 2 reduces to the encoding scheme typically employed in BERT for text paragraphs, where the [VAL_SEP] corresponds to the [SEP] token.

**Key-Aggregation:** Once we create *key-representations* we utilize them for an end-task. We encode the key-representations using the same model architecture as the TVM $TextEncoder$ but do not use positional embeddings since we encode a *set* of position-invariant key representations. Note that the weights of the KA are randomly initialized. This network is directly optimized for an end-task.

$$\mathrm{Embd}(\mathcal{J}) = \mathrm{KA}\left(\{\mathrm{KR}(k) \mid k \in \mathcal{K}\}\right) \tag{3}$$

**Key-centric vs. Record-centric Representation:** As an alternative to the *key-centric* representation used by the TVM, one could construct a *record-centric* view to model the sequence (de Souza Pereira Moreira et al., 2021; Padhi et al., 2021). Instead of modeling the evolution of *keys* in a semi-structured object sequence using $V_k$ for each key, one could treat the sequence as a series of $J_i$ (Figure 1(f)). However, the record-centric representation requires the network to compress information present in multiple keys and values of a record ($J_i$) which can create an information bottleneck for the downstream task. We compare and contrast these alternative views in Section 3 and Section 5.

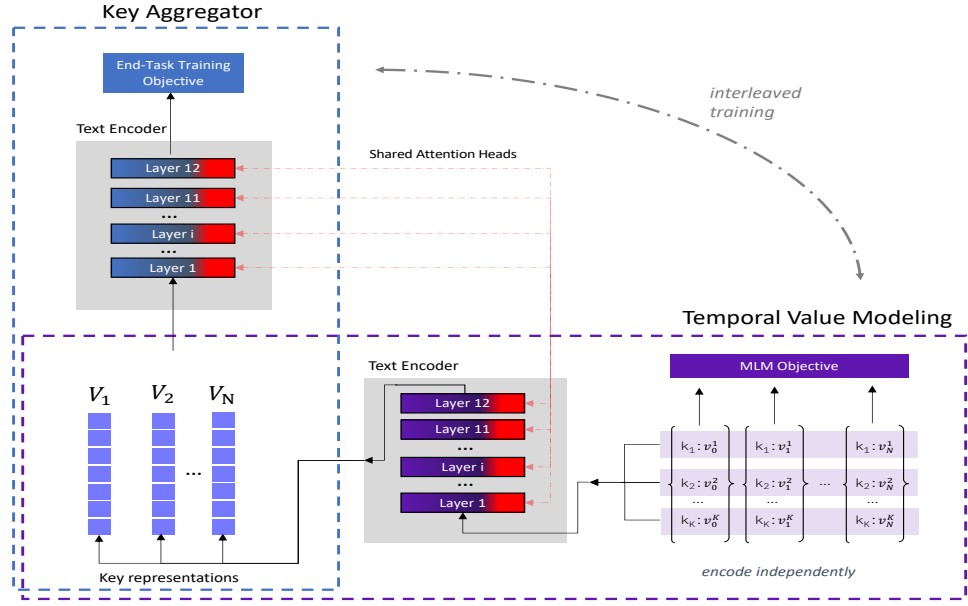

Figure 2: The TVM-KA network architecture consisting of a set of shared attention heads (weights) between the Temporal Value Modeler and the Key Aggregator. Each key is encoded independently to create a corresponding key representation.

**Challenges of Scalable Training:** The training of the hierarchical two-part network, which first obtains the *key-representations* (Eq. 2) and then the structured object sequence representation (Eq. 3), could be done *end-to-end* where the network parameters are directly trained for the downstream task. However, end-to-end training of the hierarchical two-part network is often difficult due to the constraints imposed by the limited GPU memory. The GPU memory usage is affected by two factors: (1) the length of the semi-structured object sequence; and (2) the number of keys in an object. The end-to-end model architecture operating over a batch of $\mathcal{J}$ sequences would exceed the memory of most commodity GPUs. By a conservative estimate, even for $n = 11$ and $N = 512$, a typical 120M parameter model would exceed 40GB RAM limit with a batch-size of 2. To address this, we use an iterative training paradigm, described below, which interleaves the training of the TVM and KA components by relying on attention heads that are shared between the two components.

**Sharing Attention Heads:** Recall that the TVM network first creates a representation for each key by attending to the values that occur in the sequence for each of them. The KA network then uses these representations to learn the end task. However, if the KA network could influence *how* these representations are created for each key, it could perhaps

help improve the performance of the KA on the downstream task. We, therefore, introduce hard-parameter sharing between the TVM and KA components. We hypothesize that by sharing a few attention heads (weights) between the two networks, the KA will be able to utilize the shared attention heads. Specifically, as training progresses and updates the parameters used in these heads, it will have an effect of adjusting the key-representations from the TVM in a way that could help improve overall end-task performance.

**Interleaved Task Training:** As mentioned above, we use an iterative task training paradigm where we interleave the training of the TVM and KA components. Note that our training paradigm is different from traditional training schedules for sequential task training where one network is fully trained before the next module, or from fine-tuning approaches where a part of the network may be initialized with a pre-trained model and additional layers of the network are initialized randomly and then updated for an end-task. We use the Masked language modeling (MLM) objective (Devlin et al., 2019) to train the TVM component and an end-task-specific objective for training the KA. The use of interleaved training, as outlined in Algorithm 1, prevents the problem of catastrophic forgetting (French, 1999; McCloskey and Cohen, 1989; Mc-Clelland et al., 1995; Kumaran et al., 2016; Ratcliff,

1990) when the KA is trained. Further, it is possible that when the TVM is trained for the first time it may rely heavily on the heads that are shared. Thus, any change to the representation from these heads could lead to poorer *key-representations* and attention sharing, and would therefore would be counter-productive. To address this problem, we apply DropHead (Zhou et al., 2020) on the shared attention heads in TVM and pre-train the model before beginning the interleaving schedule.

---

**Algorithm 1** Interleaved training

---

1: **Initialize** Temporal Value Modeler $\mathcal{M}_v$, Key Aggregator $\mathcal{M}_k$ parameters randomly.
2: Prepare the dataset $D_v$ consisting of value sequences
3: **for** i = 1,2,...,p **do**
4:   ▷ TVM training
5:   Update TVM $\mathcal{M}_v$ model parameters with MLM objective on $D_v$.
6:   Prepare the dataset $D_k$ consisting of key-representations $\text{KR}_k$ as per Eq. 2
7:   ▷ KA training
8:   Update Key Aggregator $\mathcal{M}_k$ model parameters with cross-entropy loss for downstream task.
9: **end for**

---

## 3 Experiments

Our experiments are designed to answer the following questions: (i) How helpful is the TVM-KA architecture over the baseline that involves flattening semi-structured object sequences? (ii) How does the model compare to existing approaches based on record-centric representations? (iii) How important is the use of shared attention heads for fine-tuning KA? (iv) Does the interleaved training procedure help train the network effectively?

### 3.1 Data

We experiment using two application/cloud logs datasets and one e-commerce purchase history dataset. The first logs dataset, referred to as 'Cloud Service Logs,' is an internal dataset consisting of interaction traces typically used for product usage analysis. We also use the publicly available LogHub (He et al., 2020) dataset, comprising system log messages from the Hadoop distributed file system, and a publicly available e-commerce dataset, which consists of product purchase information (Stanley et al., 2017). We include additional details about the datasets in the appendix.

**Cloud Service Logs Data – Application event traces from a large cloud provider:** In the Cloud Service Logs dataset, application event

traces are logged by the cloud provider website. Event types include login, browsing, account creation/maintenance/update, UI navigation, search, service creation/deletion, app interactions, and others. Each event has an associated payload that provides context around the event. Our raw data is a snapshot of application event traces spanning 3 months and comprising about 450M events, from which we build our user sessions. A user session is essentially a temporal sequence of event traces for that user. While the raw data has over 60 keys in each event, we experiment with a smaller set of manually selected 11 keys, so that existing approaches and baselines can be meaningfully used for comparison. We constructed user sessions for 100k users. The application events corresponding to 1) plan upgrade, and 2) opening chatbot (to seek help) are considered as *milestone events*. These milestone events are chosen to represent revenue generation and user experience, respectively. The case of 3) no milestone event occurring is treated as third class. From the traces, temporal sequences of 300 events are extracted to predict if a milestone (or no-milestone) event will occur in next 50 time steps. We report Macro F1-Score, as the dataset exhibits class imbalance.

**Instacart eCommerce Data:** The publicly available Instacart dataset[6] contains 3 million grocery purchase orders of nearly $200,000$ users of the application. Each order consists of multiple products and each structured object associated with a product contains meta-data such as day of the week, product category, department, aisle, etc. We re-process this dataset to create sequences of product purchases and evaluate models on the task of the next product prediction. We predict the product name given the sequence of product orders,[7] which is effectively a *classification task* over a universe of 3212 products. Existing work on this dataset has focused on a simpler binary prediction task where models are asked to predict if a particular item is likely to be purchased again.[8]

**LogHub Data:** We use the HDFS-1 from LogHub (He et al., 2020) for the log anomaly detection task. As the dataset originally consisted of lines of log messages, we use the Drain log parser

---

[6]https://tech.instacart.com/3-million-instacart-orders-open-sourced-d40d29ead6f2
[7]We use the complete structured object.
[8]https://www.kaggle.com/competitions/instacart-market-basket-analysis/leaderboard

| Dataset | Train | Dev | Test | # Classes | Task | # Keys | # Time Steps | |
|---|---|---|---|---|---|---|---|---|
| | | | | | | | Median | Maximum |
| Cloud Service Logs | 12,833 | 1,605 | 1,604 | 3 | Milestone Prediction | 11 | 112 (17061) | 300 (177340) |
| LogHub | 402,542 | 57,506 | 115,012 | 2 | Anomaly Detection | 46 | 19 (1176) | 298 (18530) |
| Instacart | 780,003 | 97,501 | 97,500 | 3,212 | Next Product Prediction | 10 | 134 (9842) | 3598 (267025) |

Table 1: Dataset Statistics including the median and maximum length of sequences reported in number of time-steps. Values in parentheses report the sequence length after sub-word tokenization using the BERT tokenizer.

(He et al., 2017) to identify 48 log templates. Using a semi-automated approach, we assign key names to the value slots of the templates. Thus, each log line is converted to a structured object with 46 key-value pairs. The original dataset splits the log lines into *blocks,* and the binary prediction task is to predict whether a particular block is *anomalous*. The dataset is highly imbalanced, with around 3% of the instances belonging to the anomalous class. We therefore report F1-Score for the anomalous class.

## 3.2 Encoders

**Baselines:** We flatten each key-value pair in a structured object and encode them with special markers indicating boundaries for objects and timesteps. We fine-tune the pre-trained encoders for each downstream task and report their performance. We experiment with BERT (Devlin et al., 2019) and Longformer (Beltagy et al., 2020) as the pre-trained encoders.

We also compare our model with popular approaches for creating record-centric representations. These approaches first obtain the representations for each object $J_i$, and then feed them to an inter-object transformer to create the representation of the whole sequence of objects. We experiment with three popular methods, where each object representation is obtained from the key-value pair representations by 1) point-wise summation, 2) concatenation then project-down (Mizrachi and Levin, 2019; de Souza Pereira Moreira et al., 2021), and 3) self-attention then averaging (Zhang et al., 2019; Gu et al., 2021). Note that the record-centric baseline can be viewed as an adaptation of the TabBERT model originally designed for tabular data (Padhi et al., 2021). Padhi et al. (2021) use a field transformer (a transformer encoder similar to BERT) to encode all rows in the table independently to obtain row embeddings. These row embeddings are later passed through another encoder to obtain a representation for the entire table. We use self-attention instead of a field transformer to encode

the key-value pairs of a JSON object.

Pre-trained models for tokens cannot be meaningfully used to initialize record-centric baselines as these create sequences over sets of key-value pairs. On the other hand, identifying a custom pre-training objective for baselines is challenging as the downstream task of predicting the next event/product involves predicting a value for a very specific key at a future time step (which is very similar to an objective one might think of for pre-training). Hence, we randomly initialize the weights of the record-centric models.

**Encoders for TVM-KA:** One of the advantages of the TVM-KA architecture is that it is agnostic to the choice of the encoder. We employ the same encoder architectures used in our baselines to enable a direct performance comparison. Recall that the TVM module and KA module share attention heads to facilitate sharing of information between them. To pre-train the TVM, we mask 15% of the tokens in every Value Sequence, and the objective is to predict the masked tokens. We do not mask *value-separator* and *key aggregator* tokens; we only mask the values. Details on hyper-parameter tuning and the iterative training schedule are in the appendix.

## 3.3 Results

Table 2 reports the primary results from our experiments. We include the results on three datasets and for each dataset, we report the overall performance along with the performance of the models on slices of the dataset where the length of the sequence is greater than the median length for that dataset.

**Comparison with Flattened encoding:** As seen in the 'overall' scores for each dataset in Table 2, flattening (first four rows) yields a significantly lower performance compared to our approach involving the use of interleaved training for TVM-KA (last row). For instance, on the cloud service logs, there's an increase of 3.9%-58.5% compared to flattened encodings in macro F1 scores. Similar

| | Configuration | Cloud Service Logs (Macro F1) | | Instacart (Recall@10) | | Loghub (Binary F1-Score) | |
|---|---|---|---|---|---|---|---|
| | | L >Median (50%) | Overall | L >Median (50%) | Overall | L >Median (51.37%) | Overall |
| Flattened Encoding (BERT) ([Devlin et al., 2019](#)) | Random (bert-base-uncased) | 49.55 | 50.22 | 9.6 | 9.4 | 0.00 | 53.62 |
| | Pre-Trained (bert-base-uncased) | 77.30 | 74.77 | 20.10 | 18.70 | 23.32 | 61.63 |
| | Pre-Trained (bert-large-uncased) | 80.06 | 76.59 | 21.07 | 19.11 | 58.30 | 75.86 |
| Flattened Encoding (Longformer) ([Beltagy et al., 2020](#)) | Pre-Trained | 75.83 | 73.71 | 16.94 | 16.08 | 97.71 | 98.54 |
| Record-centric Representation ([de Souza Pereira Moreira et al., 2021](#)) ([Gu et al., 2021](#)) | Summation | 78.97 | 77.33 | 7.11 | 5.10 | 99.22 | 99.51 |
| | Concat | 77.76 | 75.99 | 7.18 | 5.11 | **99.29** | **99.57** |
| | Self-Attention | 79.18 | 77.73 | 7.98 | 6.34 | 99.08 | 99.42 |
| TVM-KA | Joint Modeling | 80.15 | 77.68 | 17.04 | 16.0 | 46.78 | 70.72 |
| | No Interleaving | 73.19 | 73.22 | 18.32 | 17.5 | 99.05 | 98.64 |
| | Interleaving | **81.26** | **79.60** | **23.44** | **22.54** | 98.79 | 99.32 |

Table 2: Comparison of TVM-KA model with the baseline approaches on various datasets. In our proposed approach, both TVM and KA components have the same architecture and the number of parameters as *bert-base-uncased*. We additionally report results on a subset of the test set whose sequence length (L) (post-tokenization) is greater than the median length for each dataset. The values in parenthesis indicate the percentage number of instances where the length of a sequence is greater than the median sequence length. The results from our approach are statistically significant with respect to all other approaches on both cloud service logs and instacart datasets (p-value < 0.03).

trends are reported on the Instacart dataset. We find that on the LogHub dataset, the Longformer model ([Beltagy et al., 2020](#)) is able to obtain a comparable performance (98.54 vs 99.32).

**Comparison with Record-centric representations:** Unlike flattened encoding, the record-centric representation does not suffer from the modeling limitations associated with the maximum sequence length limit. These representations can encode sequences in their entirety, since most datasets have fewer than 300 objects (time-steps), and the sequence length is equal to the number of time steps. However, the record-centric view may not adequately model the dependencies between values of the same key across different time steps. We find that our approach outperforms all record-centric representation baselines on the Cloud Service Logs dataset as well as the Instacart dataset. As before, the performance of TVM-KA on the LogHub data is similar to that of different record-centric baselines (99.32 vs 99.57) - this perhaps indicates that the prediction task is relatively simpler on this dataset.

While the poor performance of record-centric baselines over flattened encoding on the Instacart dataset may appear counter-intuitive at first glance, the reason this can happen is that the record-centric approach creates a combined representation for each time-step before it processes a sequence (See Figure 1). This combined representation can be lossy and it does not allow interaction across keys or time steps, which the flattening baseline permits (thanks to $n^2$ attention across each token). Further, not only is the number of classes for prediction in the Instacart data in the thousands (while it is <5 for the other datasets), but it also exhibits longer sequences than other datasets, all of which are factors that likely contribute to this result.

**Importance of interleaved training:** As seen in the last two rows of Table 2, using our interleaving training method outperforms training the model with no interleaving. This supports our hypothesis that sharing a few attention heads helps the TVM-KA model uncover better key-representations for the downstream task as training progresses. We illustrate the increase in performance with each stage of interleaving on all three datasets in the appendix.

**Joint Modeling vs Interleaved Training of TVM-KA:** In the joint modeling approach, we do not use shared attention heads and instead train the TVM and KA networks end-to-end (jointly). We observe that *joint modeling* performs poorly compared to our interleaving approach. We found this surprising as we had expected it to be at par

with our approach when the joint models fit in memory.[9] We hypothesize that by interleaving and sharing attention heads between TVM and KA, the fine-tuning of the KA on the downstream task introduces a task-specific bias to help improve the key-representations. This in-turn, benefits the pre-trained representations of the TVM via shared-attention weights and further improves performance in the next round of training. In the absence of this bias, the joint model perhaps converges at an alternative minima that is not as good.

**Effect of varying the number of shared attention heads:** In general, we observe that sharing of $4$ and $6$ attention heads helps the most. Sharing too few or too many attention heads results in an average drop of 2%-43% in performance. We include further details in the appendix.

**Effect of parameter size/model capacity:** We investigate if the model capacity could be a bottleneck for the approaches based on flattened representations. We fine-tune a *bert-large-uncased* model, which has $3x$ the parameters of the *bert-base-uncased* model and approximately $1.5x$ the parameters of the TVM-KA network. We find that our model performs better, suggesting that the improved performance is primarily due to the value-aware key representations and the model's resultant ability to accommodate longer sequences due to the decoupling of keys.

## 4 Related Work

Our work is related to several areas of research.

**Modeling Tabular and Timeseries Data:** As we explicitly model the value sequence for each key, the data object we work with is reminiscent of tabular data where each row is a time-step and each column comprises the values for a particular field. On the surface, the modeling of data may appear related, but the actual tasks and models developed for tasks on tabular data cannot be applied to semi-structured object sequences. This is because the work on modeling textual tabular data often involves developing specialized models focused on retrieving information from cells (Zayats et al., 2021; Wang et al., 2021; Iida et al., 2021; Yang et al., 2022), multi-hop reasoning across information in different cells across parts of the table (Chen et al., 2021; Zhao et al., 2022a), combining

---

[9]We trained our joint models on 80GB A100 GPUs.

information present in tables and unstructured text for information seeking tasks (Li et al., 2021; Zhu et al., 2021; Zayats et al., 2021; Luetto et al., 2023; Cholakov and Kolev, 2022), etc. In addition, work on modeling time-series tabular data has focused on numerical data (Zhou et al., 2021; Zerveas et al., 2021; Zhao et al., 2022b) with purpose-built task-specific architectures that cannot be easily adapted to other tasks (Wu et al., 2021; Padhi et al., 2021).

**Modeling Temporal Graph Sequences and Recommender Systems:** Our approach is also related to a rich body of work on modeling temporal graphs and recommendation systems (Xu et al., 2021b; Grigsby et al., 2021; de Souza Pereira Moreira et al., 2021). Temporal graph evolution problems involve constructing representations to enable tasks such as link prediction (Sankar et al., 2020; Xu et al., 2021a), item recommendation in user sessions (Hsu and te Li, 2021), answering queries on graphs and sequences (Saxena et al., 2022), classifying graph instances in a sequence,(Xu et al., 2021a,b) etc. Our findings suggest that such approaches do not scale for long sequences for the tasks we experimented with. However, our record-centric model baselines (de Souza Pereira Moreira et al., 2021; Mizrachi and Levin, 2019) are similar in approach to these methods.

**Parameter sharing in neural network models:** Deep neural networks are usually trained to tackle different tasks in isolation. Networks that cater to multiple related tasks (multi-task neural networks) seek to improve generalization and process data efficiently through parameter sharing and joint learning. Traditional hard-parameter sharing uses the same initial layers and splits the network into task-specific branches at an ad hoc point (Guo et al., 2018; Lu et al., 2017). On the other hand, soft-parameter sharing shares features via a set of task-specific networks (Liu et al., 2019; Maninis et al., 2019). More recently adaptive sharing approaches have been proposed that decide what parameters to share across tasks to achieve the best performance (Vandenhende et al., 2019; Sun et al., 2020). The parameter sharing utilized in this work is different from the aforementioned approaches, as we share some attention head weights between the two networks (as compared to shared layers), in a way that could improve overall end-task performance.

## 5 Discussion and Conclusion

In this paper, we have presented a two-part encoder to model structured object sequences. The choice of a key-centric representation enables us to encode larger objects as well as long sequences. Our experiments show that by using the two-part TVM-KA architecture we are able to inject downstream task information into the temporal value modeler network to generate key representations that are more relevant.

We additionally present a novel interleaving scheme to train our two-part encoder. We induce task bias into the model by sharing attention heads between the Temporal Value Modeler and Key Aggregator components. Our proposed approach outperforms both the baseline approaches that flatten structured object sequences and those based on record-centric representations.

## Limitations

The benefits of our approach are best highlighted in datasets that have a large number of keys in each object when sequences are long and have a challenging prediction task. In such cases, the joint modeling of TVM-KA becomes too big to fit in memory and record-centric approaches suffer a lot of deterioration due to lossy record-representations. In our experiments, the Instacart data exhibits some of these characteristics and we see a significant improvement in performance as compared to record-centric baselines.

We note that the key-centric representation does not allow the model to support tasks such as sequence tagging of the structured objects. Nor does it allow to modeling of graph sequences effectively, as it does not use a global view of the structured objects. Thus, it may not be able to learn patterns *across fields* at different time steps. For such tasks, a record-centric representation is perhaps more helpful. Both key-centric and record-centric representations have their strengths and weaknesses, and the choice should be made with the downstream task in mind. Further, while our experiments have been reported on real-world datasets they do not represent the full spectrum of existing sequence modeling tasks.

## Acknowledgements

We would like to thank Vignesh P for his work during internship at IBM and helping us with the processing of LogHub data.

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

# A  Dataset Details

We provide additional details about the datasets used in our experiments.

## A.1  Cloud Service Logs Data: Application event traces from a large cloud provider

In the Cloud Service Logs (CSL) dataset, application event traces are logged in the cloud provider website. Each user is assigned a unique identifier. Event types range from login, browsing, account creation, account maintenance, account update, UI navigation, search, service creation, service deletion, app interactions, among several others. We have about 638 unique event types. Each event has an associated payload that provides context around the event. For example, if a user performed a search, the payload captures the search query and the page where the search was performed. If a user interacted with a service, the payload captures the service ID and action performed on the service, among other information.

Our raw data is a snapshot of application event traces spanning 3 months comprising about 450M events. Using these, we build our user sessions. A user session is essentially a temporal sequence of event traces for that user. If there is a difference of greater than 15 minutes between two consecutive events, we break the session. We constructed user sessions for 100k users. The application events corresponding to 1) plan upgrade, and 2) opening chatbot (to seek help) are considered as *milestone events*. These milestone events are chosen as they represent revenue generation and user experience, respectively. The case of no milestone event occurring is treated as the third class.

From the traces, we identify user sessions containing any of the aforementioned milestone events. We consider the temporal sequences of events 350 time-steps before the milestone event occurs. To construct the data, we consider the sequence of events till 300 time-steps and the task is to predict

if a milestone (or no milestone) event will occur in the next 50 time steps.

## A.2 Instacart eCommerce Data

The publicly available Instacart dataset[10] contains 3 million grocery purchase orders of nearly $200,000$ users of the application. Each order consists of multiple products and each structured object associated with a product contains meta-data such as the day of the week, the product category, department, aisle, etc. We reprocess this dataset to create sequences of product purchases and evaluate models on the task of the next product prediction. We create variable-length training instances from each user's order history by sampling between 50 to 200 previous product purchases for a certain target product. Additionally, we only sample a training instance if the target product has been ordered at least 50 times across users.

As per our task formulation, we predict the product name given the sequence of product orders[11], which is effectively a classification task over a universe of 3212 products. Existing work on this dataset has focused on a simpler binary prediction task where models are asked to predict whether a particular item is likely to be purchased again.[12]

### A.2.1 LogHub Data

HDFS-1 from LogHub (He et al., 2020) is utilized for the log anomaly detection task. The dataset consists of log lines. A sample log message is shown below:

```
081109 203519 29 INFO dfs.
    FSNamesystem: BLOCK*
    NameSystem.addStoredBlock:
    blockMap updated:
    10.250.10.6:50010 is added to
    blk_−1608999687919862906 size
    91178
```

We now convert the log message to a JSON object. We utilize Drain log parser (He et al., 2017) to extract the static template, dynamic variables, and header information from log messages. We obtain around 48 templates. All the log messages fall into one of the 48 templates. In a semi-automated fashion, we define keys for the templates. We populate key names with the value slots of the templates.

Thus, each log line is converted to a structured object. The log message after conversion to a JSON object would look as follows,

```
{
    "status": "addStoredBLock:
    Blockmap updated",
    "port": "10.250.10.6:50010",
    "block_ID": "blk_-160899968791986290
        6",
    "size": "91178",
    "LineId": "11",
    "Date": "81109",
    "Time": "203519",
    "Pid": "29",
    "Level": "INFO",
    "Component": "dfs.FSNamesystem",
    "EventId": "5d5de21c"
}
```

A *block* consists of a sequence of such structured objects. The task is to classify the given block as anomalous or not.

## B Hyperparameters and Training schedule

We perform a grid search for the learning rate and batch size for all the models in our experiments. We select the hyper-parameter configuration which gives the best validation set performance on each dataset's metric. We now list the range of values considered for each hyper-parameter.

- **Learning Rate:** $\{1e^{-4}, 3e^{-4}, 5e^{-4}, 1e^{-5}, 3e^{-5}, 5e^{-5}, 1e^{-6}, 3e^{-6}, 5e^{-6}\}$,

- **Batch Size:** $2, 4, 8, 16, 32$

- **Shared Heads (p):** $2, 4, 6, 8$

The drophead mechanism is only activated during TVM training, with drophead probability set to 0.2.

For all the baseline models, we train till convergence. For the TVM training in the first iteration, we vary learning rates and observe model convergence (in terms of train and dev loss) after a fixed $100K$ steps. The best learning rate for TVM is identified from this exercise. A similar approach is used for identifying the best learning rate for the KA training stage too, albeit for a smaller number of update steps. In the first iteration, TVM training is done for 1 epoch. Then we proceed with the interleaving step. We alternate between TVM

---

[10]https://tech.instacart.com/3-million-instacart-orders-open-sourced-d40d29ead6f2

[11]We use the complete structured object.

[12]https://www.kaggle.com/competitions/instacart-market-basket-analysis/leaderboard

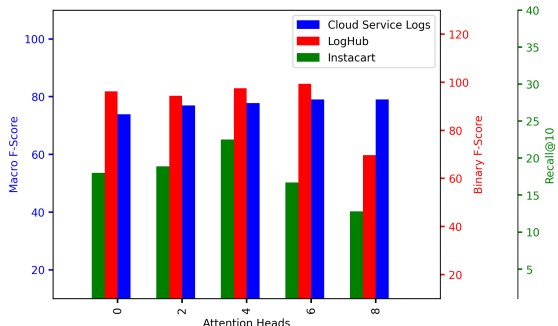

Figure 3: Effect of sharing different numbers of attention heads between TVM and KA

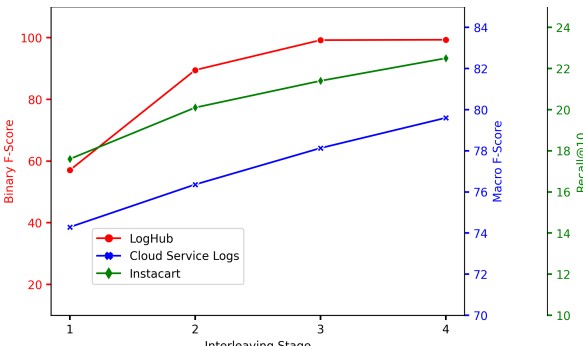

Figure 4: TVM-KA model performance for each interleaving stage. The red, green, and blue lines, along with their respective colored y-axes, indicate the performance of the Loghub, Instacart, and Cloud Service Logs datasets, respectively.

training and KA training with their number of training steps in 2:1 proportion. For the cloud service logs dataset, the number of TVM training steps is 50K and the number of KA training steps is around 100K.

## C Class-wise Results on Cloud Service Logs

Table 3 reports the detailed class-wise results on the Cloud Service Logs dataset. As seen in the 'Macro F1' score column in the Table 3, flattening (first four rows) yields a significantly lower performance compared to our approach involving the use of interleaved training for TVM-KA (last row). Specifically, the flattened encoding with randomly initialized BERT model suffers the most on the `Open Chatbot` class. We believe that semantic understanding of the event names is crucial for identifying the sequences leading to `Open Chatbot` milestone event.

In general, we observe improvements from our TVM-KA approach on all class labels compared to the baseline models. This indicates the performance gain from our model is not due to improvements in a single class or subset of classes, but, on all the classes present in our dataset.

## D Influence of shared attention heads

We use BERT encoder (Devlin et al., 2019) to model both TVM and KA components in our model. This allows us to share attention heads between the TVM and KA components. *bert-base-uncased* has 12 attention heads at each encoder layer. We experiment with sharing 0, 2, 4, 6, 8 attention heads between TVM and KA components.

Figure 3 presents the performance of our TVM-KA approach with different numbers of shared attention heads between TVM and KA components.

In general, we observe that sharing of 4 and 6 attention heads helps the most. Either not sharing any attention heads or sharing more than that results in poorer performance.

## E Training schedule

As shown in Table 2, using our interleaving training method outperforms training the model with no interleaving. This supports our hypothesis that sharing a few attention heads helps the TVM-KA model uncover better key-representations for the downstream task as training progresses. Figure 4 illustrates the increase in performance with each stage of interleaving on all three datasets.

| | Comments | Macro F1 | Micro F1 | Browsing/ Upgrade Account | No MileStone | Open Chatbot |
|---|---|---|---|---|---|---|
| Flattened Encoding (BERT) (Devlin et al., 2019) | Random (bert-base-uncased) | 50.22 | 72.38 | 70.14 | 80.50 | 0.0 |
| | Pre-Trained (bert-base-uncased) | 74.77 | 80.31 | 79.39 | 84.12 | 60.79 |
| | Pre-Trained (bert-large-uncased) | 76.59 | 81.57 | 80.72 | 85.13 | **63.93** |
| Flattened Encoding (Longformer) (Beltagy et al., 2020) | Pre-Trained | 73.71 | 79.91 | 79.31 | 84.26 | 57.57 |
| Record-centric Representation (de Souza Pereira Moreira et al., 2021) (Gu et al., 2021) | Summation | 77.33 | 85.66 | 84.91 | 90.59 | 56.50 |
| | Concat | 75.99 | 85.12 | 84.47 | 90.15 | 53.36 |
| | Self-Attention | 77.73 | 85.66 | 84.78 | 90.48 | 57.92 |
| TVM-KA | Joint Modeling | 69.61 | 80.54 | 80.04 | 86.82 | 41.97 |
| | No Interleaving | 73.22 | 84.14 | 83.03 | 90.32 | 46.31 |
| | **Interleaving** | **79.60** | **86.49** | **85.47** | **91.24** | 62.11 |

Table 3: Comparison of TVM-KA model with the baseline approaches on Cloud Service Logs datasets. In our proposed approach, both TVM and KA components have the same architecture and the number of parameters as *bert-base-uncased*. We additionally report class-wise results. The results from our approach are statistically significant with respect to all other approaches (p-value $< 0.03$).