# OpenReview forum: "Semi-Structured Object Sequence Encoders"
_EMNLP/2023/Conference — EMNLP 2023 Findings_

### Official Review · Reviewer_ZUvN · 2023-08-05

**Soundness:** 3

**Excitement:**

4: Strong: This paper deepens the understanding of some phenomenon or lowers the barriers to an existing research direction.

**Paper Topic And Main Contributions:**

Brief Summary: The paper tackles the task of how to learn over semi-structured sequences. While transformers have been used for text-sequences, here, the task is to learn series of key-value pairs which is a common data structure with many real-world use cases. The authors point out a trivial flattening of the data structure is sub-optimal due to very large sequence length. Instead, the authors explore key-centric and record centric representation. To this end, the authors propose TVM (temporal modeler) for the time-series, and KA (key aggregator) for key-centric representation, and further suggest using shared heads and training with both key-centric and record-centric representations (interleave training).

Experiments are performed on three datasets, Cloud Service Logs (internal), LogHub, and Instacart, the latter of the two being public. The authors find their proposed method (TVM-KA) outperforms competitive baselines on each dataset.

**Questions For The Authors:**

Please see previous sections (reasons to reject).

**Reasons To Accept:**

Pros:

1. The topic is quite interesting given the abundance of semi-structured representation (esp in tabular data) I can see lot of use cases for end-user. Simple flattening of the data is indeed sub-optimal. As such, the paper is well motivated.

2. The proposed method makes sense, and performing both key-aggregation and temporal modeling in the training is helpful.


**Reasons To Reject:**

Cons:

1. The main baseline considered is the flattened encoding (BERT) which handles 2/3 datasets comparatively well obtaining in 2-5 points of TVM-KA. Only on LogHub, it performs significantly worse which is due to sequence length limitation, but another flattened encoder (Longformer) is competitive. As such, the central thesis of using key-centric and record-centric over flattened encoding is not fully supported (though competitive performance is obtained by different models).

2. The authors should also show results on text-only benchmarks (like GLUE used in BERT paper), to see if TVM-KA degrades the performance on learning pure text representation (represented like in Fig 1 c). This would suggest if the TVM-KA is flexible enough to tackle very diverse forms of data.

3. A key-centric baseline should be explored. Very naively, one could use same models as for record centric representation but with k and t changed (in Fig 1f).

4. It is not intuitive as to why interleaving works better than simple joint training, and the hypothesis in L477-486 is not convincing to me. Is it possible that it is a case of simply weighting the losses in the joint setup?

5. (Minor) Some visualizations of the outputs and failure cases would be useful (in the supplementary)

**Reproducibility:**

2: Would be hard pressed to reproduce the results. The contribution depends on data that are simply not available outside the author's institution or consortium; not enough details are provided.

**Reviewer Confidence:**

3: Pretty sure, but there's a chance I missed something. Although I have a good feel for this area in general, I did not carefully check the paper's details, e.g., the math, experimental design, or novelty.

---

> ### Author Rebuttal · Authors · 2023-08-29
>
> We thank the reviewer for their feedback and for rating our paper strong and well motivated. We include a response to the comments made by the reviewer below:
>
> **The main baseline considered is the flattened encoding (BERT) which handles 2/3 datasets comparatively well obtaining in 2-5 points of TVM-KA. Only on LogHub, it performs significantly worse which is due to sequence length limitation, but another flattened encoder (Longformer) is competitive. As such, the central thesis of using key-centric and record-centric over flattened encoding is not fully supported (though competitive performance is obtained by different models).**
>
> As the reviewer correctly highlights, as compared to BERT we have gains varying from 4% to 23% depending on the dataset but we likely to respectfully highlight that they are statistically significant and the performance is consistent across datasets. Further, the performance in datasets such as Instacart has an F1 of ~20-23 where gains of 2-4 points are important due the challenging nature of the task. Unfortunately datasets are difficult to source for this challenging problem and we hope the reviewer can appreciate we have tried to adapt existing publicly available datasets to demonstrate our work.
>
> **The authors should also show results on text-only benchmarks (like GLUE used in BERT paper), to see if TVM-KA degrades the performance on learning pure text representation (represented like in Fig 1 c). This would suggest if the TVM-KA is flexible enough to tackle very diverse forms of data.**
>
> As discussed in the main paper, for text only representations, the TVM network is effectively the base model (and can be any existing pre-trained model). In such a setup the use of the KA network is unnecessary - as an example, if the TVM model is based on BERT and lets say our NLP task is classification, the KA network will only need to learn to copy the CLS embedding from TVM.
>
> **A key-centric baseline should be explored. Very naively, one could use same models as for record centric representation but with k and t changed (in Fig 1f).**
>
> We apologize that we are unsure we understand what the reviewer is suggesting.  Could you please clarify? Our flattened baseline, uses a series of key:value pairs in sequence as shown in Figure 1(d), our record-centric baseline aggregates all key-values within an object before encoding it as a sequence and key-centric approach models values of each key across time-step before aggregating them for use in an end-task.
>
> **It is not intuitive as to why interleaving works better than simple joint training, and the hypothesis in L477-486 is not convincing to me. Is it possible that it is a case of simply weighting the losses in the joint setup? Some visualizations of the outputs and failure cases would be useful (in the supplementary)**
>
> We agree that analyzing the effect of shared attentions is difficult but we have tried to empirically demonstrate that in our work with rigour. Our hypothesis is perhaps best understood by keeping in mind the performance of no-interleaved training (when shared attention-heads are not used - Table 2) as well as how the performance of the model improves with each stage of interleaved training (Please refer to Figure 4 in the appendix).
>
> We thank the reviewer for the suggestion about include visualizations -- we will include them in the supplementary text.
>
> **Reproducibility**
>
> We release all code and we have experimented with two publicly available datasets. We will be releasing our data processing scripts as well
>
> We hope we were able to satisfactorily address all queries that the reviewer had and we request the reviewer to account for this rebuttal in their final scores.

---

### Official Review · Reviewer_sSHy · 2023-08-05

**Soundness:** 3

**Excitement:**

4: Strong: This paper deepens the understanding of some phenomenon or lowers the barriers to an existing research direction.

**Paper Topic And Main Contributions:**

Authors treat semi-structured object sequences as key-centric representations by computing a representation for each set of keys separately along the time axis and then aggregating them using self-attention for an overall representation.

**Questions For The Authors:**

Please refer to the previous section

**Reasons To Accept:**

1. Compared to flatten or record-centered approaches, separating each key and encoding each sequence of the key separately before aggregating them together makes the collection of information more efficient and capture the correlation in a better way.
2. The paper is well structured and the figure/algorithm is clear. We can see improvement in 2/3 datasets. The ablation study is well-designed and persuasive.
3. The idea of using the shared attention head is efficient and makes sure that TVM and KA models sync with each other and share the attention learned in the sequential training.


**Reasons To Reject:**

1. I would suggest more analysis discussion, such as the difference between Joint Modeling and Interleaved Training.
2. The performance is actually not that promising. In three datasets, the proposed approach only shows consistent improvement in two of them by a small margin. For the remaining one, the Record-centric approach actually works the best. What would be the kind of tabular data that would benefit from the proposed approach? Is there any proper reason why it's not working on Loghub dataset?
3. A baseline on key-centric representation would be helpful, such as simply passing it into a transformer encoder separately and aggregate them (mean/sum) for the classification.

**Reproducibility:**

3: Could reproduce the results with some difficulty. The settings of parameters are underspecified or subjectively determined; the training/evaluation data are not widely available.

**Reviewer Confidence:**

3: Pretty sure, but there's a chance I missed something. Although I have a good feel for this area in general, I did not carefully check the paper's details, e.g., the math, experimental design, or novelty.

---

> ### Author Rebuttal · Authors · 2023-08-29
>
> We thank the the reviewer for their feedback! We address all points raised by the reviewer below.
>
> **Difference between Joint Modeling and Interleaved Training.**
>
>
> In case of joint modeling we do not use shared attention heads and instead train the TVM and KA networks end-to-end (jointly). In contrast, interleaved training process first pre-trains the TVM with shared attention heads and then the KA network where TVM parameters (except shared attention heads are frozen), followed by training the TVM again where KA parameters are frozen (except shared attention heads). We refer to this process as "interleaving". Please refer to the sections between lines 2292-86 in the main paper for more details, including the motivation of the approach. We thank the reviewer for this question and we will further clarify this in the revised version of the paper.
>
> **The performance is actually not that promising. In three datasets, the proposed approach only shows consistent improvement in two of them by a small margin. For the remaining one, the Record-centric approach actually works the best. What would be the kind of tabular data that would benefit from the proposed approach? Is there any proper reason why it's not working on Loghub dataset?**
>
>
> There are three main take-aways from the results in Table 2.
>
>
> (i) When using flattened-encoding, using models with support for longer context lengths can help improve performance (eg: BERT vs Longformer)
>
>
> (ii) Introducing structure-ware representations (such as Record-centric representations with self-attention) further improve performance  (existing approaches -- our baselines).
>
>
> (iii) Key-centric representations based on the TVM-KA (our contribution) not only allow models to include significant larger input sequences, they also perform better than both flattened and record-centric representations. Note that, TVM-KA models are trained under three settings  -
>
> - (a) Joint Modeling
> - (b) Interleaved training (our contribution)
> - (c) No interleaving - i.e, one single step of TVM training followed by KA training.
>
> We respectfully highlight that *TVM-KA with interleaved training indeed performs well on all datasets including LogHub*. In the case of joint modeling, the results are lower than key-centric baselines. We hypothesize that this because on LogHub there isn't a lot of cross-key interaction across time-steps i.e, the presence of a value for a key in a sequence at one time-step correlates highly with an anomaly. However, in the joint-model training setup the network is unable to pick up on correlations due to a weak signal coming from the aggregated key-representations via the KA layer. In contrast, in case of inter-leaved training inter-leaving and sharing attention heads between TVM
> and KA, the fine-tuning of the KA on the downstream task introduces a task-specific bias to help improve the key-representations. This in-turn, benefits the pre-trained representations of the TVM via shared-attention weights and further improves performance in the next round of training. In the absence of this bias, the joint model perhaps converges at an alternative minima that is not as good. We discuss this aspect in our main paper between lines 462-492.
>
> **What kind of data would it help on?**
>
> The benefits of our approach are best highlighted in datasets that have a large number of keys in each object, when sequences are long and have a challenging prediction task . In such cases, joint modeling of TVM-KA become too big to fit in memory and record-centric approaches suffer a lot of deterioration due to lossy record-representations.  In our experiments, the Instacart data exhibits some of these characteristics and we see a significant improvement in performance as compared to record-centric baselines.
>
> We would have liked to experiment with more challenging real-world datasets but as the reviewer will appreciate, these are hard to source and we have tried to adapt existing public datasets to demonstrate our work.
>
> **Additional baseline**
>
> This baseline is effectively the TVM-KA approach with no interleaved training. Instead of a simple-sum we run self-attention which allows the model to construct richer representations and is likely a stronger baseline than the one suggested by the reviewer.
>
> We hope we were able to satisfactorily address all queries that the reviewer had and we request the reviewer to account for this rebuttal in their final scores.

---

### Official Review · Reviewer_A9A6 · 2023-08-06

**Soundness:** 2

**Excitement:**

3: Ambivalent: It has merits (e.g., it reports state-of-the-art results, the idea is nice), but there are key weaknesses (e.g., it describes incremental work), and it can significantly benefit from another round of revision. However, I won't object to accepting it if my co-reviewers champion it.

**Paper Topic And Main Contributions:**

This paper investigates a key-centric approach to encoding semi-structured object sequence. The authors compared the proposed approach with the flattening approach and record-centric approach on three datasets, demonstrating the superior performance of the key-centric approach.

**Questions For The Authors:**

Did the authors explore pre-training in the joint modeling approach? It's crucial for the authors to demonstrate that the performance boost of the interleaved approach is attributable to the interleaved training itself, rather than to pre-training.

Did the authors explore first pre-training the TVM module and then finetune the KA module?

**Reasons To Accept:**

Given that the proposed key-centric method appears as a natural choice for encoding record sequences, contrasting it with other approaches would offer valuable insights to the community.
The authors further investigates interleaving training which improves the performance of the key-centric approach.

**Reasons To Reject:**

The evaluation lacks comprehensiveness; it covers only three datasets.

The result of the record-centric approach on the Instacart dataset is not very convincing; it's even much lower than the flattening baseline.

Pre-training can have a large impact on model performance. Have the authors tried employing pre-training in the record-centric approach?

The proposed interleaved training, being more complex than other methods, might necessitate finer parameter tuning for optimal performance.

**Reproducibility:**

4: Could mostly reproduce the results, but there may be some variation because of sample variance or minor variations in their interpretation of the protocol or method.

**Reviewer Confidence:**

3: Pretty sure, but there's a chance I missed something. Although I have a good feel for this area in general, I did not carefully check the paper's details, e.g., the math, experimental design, or novelty.

---

> ### Author Rebuttal · Authors · 2023-08-28
>
> We thank the reviewer for their comments. We address the points raised by the reviewer below:
>
> **Weakness 1:The result of the record-centric approach on the Instacart dataset is not very convincing; it's even much lower than the flattening baseline.**
>
> While this may appear counter-intuitive at first glance, the reason this can happen is that the record-centric approach creates a combined representation for each time-step before it processes a sequence (See Figure 1). This combined representation can be lossy and it does not allow interaction across keys or time-steps, which the flattening baseline permits (thanks to n^2 attention across each token).  Further, not only are the number of classes for prediction in the Instacart data is in the thousands (while it is <5 for the other datasets), it also exhibits longer sequences than other datasets, all of which are factors that likely contribute to this result.
>
> **Weakness 2: Pre-training can have a large impact on model performance. Have the authors tried employing pre-training in the record-centric approach?**
>
> Pre-trained models for tokens cannot be meaningfully used to initialize record-centric baselines  as they create sequences over sets of key-value pairs.
>
> On the other hand, identifying a custom pre-training objective for this baseline is challenging as the downstream task of predicting the next event/product involves predicting a value for a very specific key at a future time step (which is very similar to an objective one might think of for pre-training).
>
> **Question 1:Did the authors explore pre-training in the joint modeling approach? It's crucial for the authors to demonstrate that the performance boost of the interleaved approach is attributable to the interleaved training itself, rather than to pre-training.**
>
> In this case of TVM-KA the value modeler can be initialized with any existing pre-trained model, however we pre-train the model ourselves because of the use of shared-attention heads (see lines: 290-297).
>
> **Question 2: Did the authors explore first pre-training the TVM module and then finetune the KA module?**
>
> Yes that setting is referred to as "No interleaving" in our experiments -- See Table 2.
>
> We hope we were able to satisfactorily address all queries that the reviewer had and request the reviewer to account for this rebuttal in their final scores.

---

### Meta-Review · Area_Chair_kXri · 2023-09-11

**Recommendation:** 4

**Metareview:**

The paper looks at the task of learning semi-structured sequences. They frame this as a task of learning  a key-value data structure.  They propose key-centric and record centric representations. They propose a temporal modeler (TVM) for the time-series, and KA (key aggregator) for key-centric representation, and further propose interleaved training to learn both representations. They evaluate and show good performance on 3 datasets.

The reviewers agree that the approach is exciting, the paper is well motivated, and ablations are persuasive. The reviewers also raise some clarifications on the evaluation and training. In particular,

&nbsp; (1) the evaluation with the flattened encoders is still competitive and hence questions whether the premise of the key-centric and record-centric representations is even necessary. The authors have addressed this in their response.

&nbsp; (2) there were also some clarifications requested regarding the interleaved training and the joint training.

The authors have responded to these questions. Perhaps there is still scope to clarify the writing, especially with regard to other possible baselines suggested by reviewers and how those might compare to the baselines in the paper. Incorporating the feedback from the reviewers should help improve the quality of the paper.

---

### Decision · Program_Chairs · 2023-10-07

**Decision:**

Accept-Findings

**Comment:**

The paper looks at the task of learning semi-structured sequences. They frame this as a task of learning  a key-value data structure.  They propose key-centric and record centric representations. They propose a temporal modeler (TVM) for the time-series, and KA (key aggregator) for key-centric representation, and further propose interleaved training to learn both representations. They evaluate and show good performance on 3 datasets.

The reviewers agree that the approach is exciting, the paper is well motivated, and ablations are persuasive. The reviewers also raise some clarifications on the evaluation and training. In particular,

&nbsp; (1) the evaluation with the flattened encoders is still competitive and hence questions whether the premise of the key-centric and record-centric representations is even necessary. The authors have addressed this in their response.

&nbsp; (2) there were also some clarifications requested regarding the interleaved training and the joint training.

The authors have responded to these questions. Perhaps there is still scope to clarify the writing, especially with regard to other possible baselines suggested by reviewers and how those might compare to the baselines in the paper. Incorporating the feedback from the reviewers should help improve the quality of the paper.